# Bracing of Pectus Carinatum in Children: Current Practices

**DOI:** 10.3390/children11040470

**Published:** 2024-04-15

**Authors:** Pavol Omanik, Sergio Bruno Sesia, Katarina Kozlikova, Veronika Schmidtova, Miroslava Funakova, Frank-Martin Haecker

**Affiliations:** 1Department of Pediatric Surgery, National Institute of Children’s Diseases, 83340 Bratislava, Slovakia; pavol.omanik@nudch.eu (P.O.); veronika.chrenkova@nudch.eu (V.S.); miroslava.funakova@nudch.eu (M.F.); 2Division of General Thoracic Surgery, Bern University Hospital (Inselspital), 3010 Bern, Switzerland; sergio.sesia@insel.ch; 3Institute of Medical Physics and Biophysics, Comenius University, 81372 Bratislava, Slovakia; katarina.kozlikova@fmed.uniba.sk; 4Department of Pediatric Surgery, Children’s Hospital of Eastern Switzerland, 9000 St. Gallen, Switzerland

**Keywords:** pectus carinatum, CWIG, survey, compressive bracing, children

## Abstract

Background: Although effective, compressive orthotic bracing (COB) in children with pectus carinatum is still not standardized. This study has aimed to analyze current practices amongst members of the Chest Wall International Group (CWIG). Methods: A web-based questionnaire was mailed to all CWIG members at 208 departments. It included 30 questions regarding diagnostic work-up, age for COB indication, type of COB used, daily wearing time, treatment duration, complications, and recurrence rate. Results: Members from 44 departments have responded (institutional response rate 21.2%). A total of 93% consider COB as the first-line treatment for PC. A conventional COB (CC) is used in 59%, and the dynamic compression system (FMF) in 41%. The overall compliance rate is >80%. A total of 67% of responders consider COB to be indicated in patients <10 years. The actual wearing time is significantly shorter than the physician-recommended time (*p* < 0.01). FMF patients experience a significantly faster response than CC patients (*p* < 0.01). No recurrence of PC has been noted in 34%; recurrence rates of 10–30% have been noted in 61%. Conclusions: COB is the first-line treatment for PC with a high compliance rate. During puberty, the recurrence rate is high. Treatment standardization and follow-up until the end of puberty are recommended to enhance COB effectiveness.

## 1. Introduction

Affecting nearly 1 in 1000 children, pectus carinatum (PC), also called ‘pigeon chest’, is characterized by the protrusion of the sternum and adjacent costal cartilage. It is the second most common, probably inherited deformity of the anterior chest wall, occurring mostly in males, with a male/female ratio of 4:1 [1]. This deformity has been referred to as “undertreated chest wall deformity” due to infrequent referrals from primary care practitioners and probable underestimation of its frequency [2,3]. PC rarely causes cardiopulmonary symptoms, but mostly adolescent patients experience shame and embarrassment resulting in low self-confidence and therefore seeking treatment [4,5].

In the past, the treatment of PC has changed substantially. For decades, the surgical technique popularized by Ravitch in 1949 was the only available option, an open technique based on extensive growth centers cartilage and bone resection [6,7]. In particular, if performed too early during childhood, this technique may cause acquired asphyxiating chondrodystrophy, a condition resulting in a narrow, bell-shaped chest and inducing a potentially life-threatening breathing situation [8].

It therefore appeared imperative to develop less invasive repair techniques for PC. Interestingly, since PC occurs more frequently than pectus excavatum in South America, surgeons from the continent became pioneers in promoting techniques such as the non-surgical external compression brace established by Haje in early 1980 and perfected by Martinez-Ferro with the dynamic compression system early in the 21st century [9,10]. Furthermore, Abramson published in 2005 his minimally invasive internal compression surgical technique consisting of the placement of a pectus bar subcutaneously above the sternum [11]. The results of compressive orthotic bracing (COB) for PC treatment have been encouraging and effective enough to make COB the first-line approach. For surgical repair, the Abramson technique is considered the new standard, and the Ravitch technique remains reserved for more complex deformities [1,11]. Although COB for PC treatment is widely applied all over the world, a standardized protocol including indication for treatment, type of bracing, duration of treatment, etc. is still lacking. The purpose of this communication is to analyze amongst Chest Wall International Group (CWIG) members whether there is a common consent concerning the above-mentioned variables and to attempt a definition of standards for the non-surgical treatment of PC.

## 2. Materials and Methods

### 2.1. Recruitment of Participants and Data Collection

The CWIG aims to advance the science and art of chest wall surgical and non-surgical procedures through research, education, and collaboration with interdisciplinary and international experts [12]. This multipurpose platform offers information, communication, and cooperation to pediatric surgeons as well as thoracic surgeons, plastic surgeons, and all other specialists concerned with disorders involving the thoracic wall. An invitation to participate was sent to all registered members of the CWIG platform (www.cwig.info, accessed on 9 April 2021). An online questionnaire was shared using direct email contact, using a survey link (Google Forms, https://docs.google.com/forms/d/1FL723k1jl2y21WDSyrBl2amJfW7zwsfl8FEivkBLOCc/edit?pli=1, accessed on 12 February 2024). The list of email addresses was handled in accordance with GDPR rules regarding secure storage. Ethical review and approval were waived for this study since no patient personal data were used.

### 2.2. The Questionnaire

The questionnaire consisted of 30 closed-ended questions focusing on COB in pectus carinatum patients: pre-treatment assessment, measurements, indications vs. contraindications, the wearing daily time of COB, complications, follow-up regime, total duration of therapy, long-term outcomes, recurrence after cessation of COB and patient’s compliance. The last question was reserved for additional comments and suggestions (Appendix A). The questionnaire was prepared by consultant pectus surgeons of two European centers of excellence and approved by the current president of the CWIG. The questionnaire was voluntary and anonymous, and it allowed skipping questions. The time needed to complete the survey did not exceed 10 min. The survey link and the possibility to participate online was active for 30 days. All completed questionnaires were included in this study, regardless of the number of questions answered.

### 2.3. Statistical Data Evaluation

Since the majority of the quantitative data were not distributed normally as indicated by the Shapiro–Wilk test, medians and ranges (maximal and minimal values) were reported. Either the Mann–Whitney U test (two groups) or the Kruskal–Wallis test (three or more groups) were used to compare medians [13].

A value of *p* < 0.05 was always considered a statistically significant difference and a value of 0.05 < *p* < 0.10 was always considered a borderline statistically significant difference. MS Excel 2019 was used for calculations and graphs. In all box and whisker plots, the median is represented as a horizontal line in the box showing the interquartile range. The diagonal cross (×) shows the arithmetic mean. If there are no outliers (empty circles (◦)), the ends of the upper and lower whiskers represent the maximum and minimum values. Otherwise, they represent the largest value not greater than the third quartile plus 1.5 fold the interquartile range, and the smallest value not less than the first quartile minus 1.5 fold the interquartile range [13].

## 3. Results

The questionnaire was sent to three hundred and thirteen colleagues from 208 departments originating from 22 countries. Forty-four completed questionnaires were retrieved and analyzed (department response rate 21.2%). The participation of the individual countries from which the answers came was as follows (alphabetically): Argentina, Australia, Belgium, Brazil, Bulgaria, Canada, France, Italy, Japan, Mexico, Netherlands, Russia, Slovakia, South Africa, South Korea, Spain, Switzerland, Taiwan, Turkey, United Kingdom, Ukraine, and the United States of America.

Each department was assigned to a region: Europe (E; 20); North America (N; 11); South America (S; 5); Asia, Africa, and Australia (A; 8). The overall extrapolated number of patients treated in participating departments approached 4750: E 2250, N 1425, S 325, A 750. The number of patients was calculated as the weighted sum of sorted data using the middle value of each class; therefore, the resulting number can be taken as a reliable one (although it seems to be high. University hospitals were the dominant type of institution (80%). PC patients were mainly treated by pediatric surgeons (59%) or by thoracic surgeons (32%); a minority of PC patients were treated in departments of pediatrics, orthopedic surgery, or plastic and reconstructive surgery. Responders were consultants (61%) and heads of department (39%). In addition to surgeons (93%), the local treatment team included a physiotherapist (55%), a psychologist (18%), an orthopedist (16%) and a clinical anthropologist (9%).

### 3.1. The Experience of the Participating Departments

The experience of the participating departments illustrates Table 1.

### 3.2. Indications and Contraindications for COB

The majority of participating departments (93%) consider COB as a first-line treatment option for PC patients. Regarding indications for COB, the questionnaire did not include the question about indication criteria. It is a generally accepted that the deformity itself is the indication for the COB treatment, as well as the patient’s need for deformity treatment. The average age of the first application is 9.5 ± 1.9 years. More than one-third of respondents (37%) apply COB in children younger than 8 years old, 30% of them consider the lowest age limit 9–10 years, 28% of respondents 11–12 years, and 5% of respondents consider the lowest age limit 13–14 years. A mutual comparison between departments divided by regions revealed no significant difference in the median age from which COB is indicated (Kruskal–Wallis test) (Figure 1).

The contraindications for COB mentioned included non-compressible chest deformity (34/43; 79%), cardiovascular diseases (5/43; 12%) and connective tissue disease (4/43; 9%). In contrast, 5/43 (12%) of respondents stated to have no contraindication to COB. (The smaller number of responses than the total of 44 reflects the fact that not all respondents answered all questions).

37/42 (88%) responders consider a failure of COB as an indication for surgical intervention. In total, 31/42 (77%) responders consider non-compressible chest deformity as COB contraindication, and 29/42 (69%) as a reason for rejection of the treatment.

A conventional compressive brace (CC) was used in 59% of departments, and the Fraire Martinez-Ferro dynamic compression system (FMF) in 41%.

### 3.3. Diagnostics and In-Treatment Protocol

Conventional photography, computer tomography, 3D optical scanning, and cardiac workup were the most frequently used methods to assess the severity of PC. Photography, clinical anthropometry, 3D scanning, and computer tomography were most frequently used to determine the improvement of the chest wall during COB (Figure 2).

### 3.4. Treatment Regime and Follow-Up

Regarding the daily wearing time of COB, a notable discrepancy appears between physician-recommended and actual (patient-confirmed) time of daily application (Figure 3). Eighty-two percent of departments recommend COB wearing even at night.

The average physicians’ recommended time of COB wearing was 14.3 ± 3.7 h in the CC group, and 16.4 ± 4.0 h in the FMF group. The average actual, patients’ confirmed time of COB wearing was 10.4 ± 3.8 h in the CC group, and 14.8 ± 5.5 h in the FMF group. A statistically significantly shorter time between the average physicians’ recommended length of daily COB application was found in the CC group compared to the FMF group (*p* < 0.01). Similarly, a statistically significantly shorter time was noticed between the average patients’ confirmed length of daily COB application in the CC compared to the FMF group (*p* < 0.01). In both the CC and FMF groups, a statistically significantly shorter time of patients’ confirmed COB wearing compared to the physicians’ recommended time was found (*p* < 0.01) (Figure 4).

Out-patient controls during COB were performed approximately once every 3 months in 17/43 (40%), more frequently than once per 3 months in 17/43 (40%), and less frequently in 9/43 (20%).

The decision when to change the treatment regime to the maintenance phase was based mainly on: patient satisfaction in 22/44 (50%), results of clinical anthropometric examination in 12/44 (27%), and results of 3D scanner image in 4/44 (9%). Termination of treatment was most often decided based on the same criteria (58%, 21%, and 5%, respectively).

The average duration of treatment with CC was 14.0 ± 4.3 months, with FMF 11.0 ± 4.0 months. The average duration of treatment was significantly longer with CC than with FMF, with the two-tailed *p* value < 0.01. At the same time, it was found that the mean age at which CC treatment was started, (9.5 ± 1.6 years), was significantly higher than the mean age at the start of FMF treatment, (7.5 ± 1.1 years) (the two-tailed *p* < 0.01) (Figure 5, Figure 6 and Figure 7).

The patients were followed-up until the end of growth in 82%. No recurrence of PC during puberty was noticed by 14/41 (34%) responders, whereas 25/41 (61%) reported recurrence rates varying from 10 to 30%, and 2/41 (5%) recurrence rate >50%. According to the Mann–Whitney U test, there is no statistically significant difference regarding the recurrence rate between the FMF and CC groups.

The most common complications during COB application were pain (35%), followed by local skin rashes (26%) and itching (19%). In total, 91% of respondents considered lack of compliance as the most common reason for treatment failure. 32/43 (74%) of the responders stated a compliance rate of COB > 80% (regarding non-compliance, see Figure 8).

The Mann–Whitney U test shows a significantly better compliance in patients treated with FMF than with CC (*p* < 0.01).

The success of treatment was confirmed by administering a treatment satisfaction questionnaire in 16/43 (37%); no questionnaire was used in 27/43 (63%).

Regarding cost recovery, COB was fully covered by health insurance in 15/44 (34%), and partly covered in 13/44 (30%); the patient had to pay for the device himself in 16/44 (37%).

## 4. Discussion

The results of this survey confirm, remarkably, CT scan as a very commonly used imaging modality to assess PC, COB as the first-line treatment for PC patients, a discrepancy between physician-recommended and actual patient-confirmed daily wearing time, and the need to follow-up the patients until the end of growth. At the same time, it shows no standard procedure regarding the method to assess PC deformity, the type of used compressive orthosis, the time of starting the therapy, the daily bracing time, and ways to evaluate treatment success. Those findings may well account for the quite high recurrence rates of 30% and more, and for the fact that non-surgical treatment of PC is still not fully covered by health insurance in the majority of countries.

Of course, the presented results have to be analyzed with great caution due to several reasons. Our findings are based on an international questionnaire study, submitted to a preselected group of experts, and therefore their conclusions cannot provide the same validity of outputs as randomized multicenter studies have. The data extrapolated from the findings in the questionnaire after their statistical processing indicate current trends, highlight common points and at the same time differences and controversies between many institutions. The results of this study do not aim to establish new algorithms, but rather to stimulate a deeper analysis of the issue, for example, through international multicenter studies, in which the validity of the results could indicate the direction towards the unification of diagnostic and therapeutic procedures.

The individual response rate of 14% and 22% institutional response rate may seem like a low value. However, the most important reason for the low response rate is that there are institutions preferring surgical treatment of PC as a method of choice (Ravitch or Abramson procedures and their modification). Based on this and on the fact that we cannot recognize the part of CWIG members, who prefer the surgical method, we can suppose that a substantial part of CWIG members did not respond to the questionnaire due to the reason of their surgical preference. In addition, the relatively low response rate must be interpreted with caution. A total of 313 colleagues was contacted, including not only pectus surgeons registered in the CWIG database, but also supportive specialists like physiotherapists and registered nurses. Finally, consultants of well known high-volume centers responded (Argentina, Belgium, Brazil, England, France, Italy, the Netherlands, South Africa, Spain, Switzerland and the USA). The centers in these countries are also those with the most experience in using the FMF.

As mentioned above, another important fact in connection with the low response rate is the fact that the CWIG is a heterogeneous society of different specialists: the ones who contribute to diagnostics, an indication of COB treatment and follow-up (pediatric surgeons, thoracic surgeons, and plastic surgeons); the others are supportive specialists (physiotherapists, physiologists, nurses, etc.), who probably did not answer the questionnaire. One could expect the answers from the first mentioned group, but we are not capable of selecting surgical specialists from the email addresses of CWIG members who participate in the follow-up of patients. Analysis of email addresses showed that individual departments have various numbers of CWIG members (at least one member, maximum of seven members). The majority of answers from individual institutions come from either the head of department or consultants which determines institutional response rate as more relevant.

COB of PC has gained great popularity due to its non-invasiveness, with no risk of anesthesia and surgery, and excellent results in compliant patients. The American Pediatric Surgical Association recommends COB as the first-line treatment for PC [3,10,14,15,16,17,18].

Although COB is mentioned in international guidelines in the therapeutic algorithm for the treatment of PC, high-quality long-term data and a standardized wearing protocol are still not available [19,20,21]. Hunt et al. confirmed the need for robust level I randomized data with a clearly standardized bracing protocol, objective measurement of outcomes, and recording of results at the end of the bracing treatment program in sufficiently powered sample sizes over a significant follow-up period [22].

PC is a disease somehow in the shadow of pectus excavatum (PE). Unlike PE, PC does not compromise the cardio-respiratory function in most patients and thus remains considered a purely cosmetic problem. Like PE, PC has a tendency of progressive aggravation with age. Thus, once the diagnosis is confirmed, patients should be actively treated [23], or at least carefully monitored. In the last two decades, the treatment of PC has seen a gradual shift away from surgical treatment (open correction and mini-invasive correction) to a non-surgical approach. Today, COB seems to be established as the standardized first choice of treatment for PC patients. Surgical repair is reserved for those patients who fail or can be expected to fail COB because of chest stiffness or patient non-compliance [24,25].

The overall extrapolated number of patients who were treated in participating departments reached 4750. The experience of individual participating departments is documented by the fact that almost 50% of them (46%) have successfully treated more than 100 patients and 25% of them have treated more than 200 patients so far. Many authors have already published their results. For comparison, de Beer et al. in a meta-analytic study, processed the results of 8 single-center studies, including 1185 patients [26]; the largest single-center studies analyzed 740 and 664 patients, respectively [23,27].

Thus far, no analysis comparing the results of CC and FMF therapy is available in the literature. The available literature comprises either single-center experience or a meta-analytic summary of one of these methods [26]. To the best of our knowledge, the present international survey is the first one comparing these two therapeutic modalities. Even if we are not able to present a detailed analysis, we observed a tendency concerning advantage vs. disadvantage of CC vs. FMF. Of course, also this finding has to be analyzed with great caution since there might be a bias. Responders were not randomized, and maybe the use of the FMF is considered more popular in comparison to CC orthesis.

Physiotherapists were part of the therapeutic team in 55%. This result confirms the important role of physiotherapy in PC treatment for postural correction since most PC patients present a kyphosis of the thoracic spine. “Pectus posture” refers to the position caused by the forward displacement of the patient’s shoulders and the development of thoracic kyphosis [28]. Abnormal posture and lack of back muscles training may worsen the deformity [23]. Even if long-term evidence of effectiveness is lacking, physical therapy for treating PC and posture is an important and often poorly recognized treatment option [19,28].

There is still no consensus on when to start COB. Two factors oppose each other when it comes to starting COB before puberty: (i) the significantly better elasticity of the anterior chest wall and the associated faster therapeutic effect, and (ii), the significantly higher risk of recurrence and of non-compliance as a result of treatment failure [27]. Available literature acknowledges successful treatment in patients aged 2–4 years [10,21,23,26,27]. In the case of recurrence, the deformity is rebraced at the onset of adolescence [10,27]. An interesting conclusion was made by the investigations of Port et al. that children who grew more while wearing COB showed greater improvement of the deformity [14]. In comparison, our results underline that the average age of the start of COB was 9.5 ± 1.9 years, with 67% of responders beginning treatment in children <10 years old.

To date, the most conventional and widely used methods for evaluating the treatment outcome and/or the severity of PC are the chest X-ray and surprisingly the CT scan [29]. Needless to say, due to the harmful radiation, a radiation-free examination method is mandatory [30]. Growing tissues are at greater risk of developing cancer after being exposed to ionizing radiation than adult tissues [31]. Pediatric patients have a long lifetime risk to develop radiation-related pathologies [32]. Radiation-free diagnostic modalities (clinical anthropometry and 3D surface scan) provide a safe, quick, valid, and easily implemented alternative to traditional irradiating assessments for pectus deformities [14,23,33]. It allows for a simple out-patient assessment of PC and appears to aid in providing an ongoing assessment tool, particularly important for therapies requiring a high degree of compliance as COB [34]. 3D body surface scanning objectifies the improvement of PC and gives positive feedback to the patient, especially valuable in PC, where changes progress slowly. This allows the patient to monitor the treatment and increase treatment adherence [35,36]. It is important to emphasize that those new imaging modalities do not provide any information such as cardiopulmonary impression and sternal torsion [34]. Standardized photographs still have an important place in diagnosis and follow-up [14,16]. The high representation of radiation-free modalities (photography, 3D surface scanner, and clinical anthropometry) was also revealed in our survey. However, it is noteworthy that 43% of departments still use a CT scan of the chest in the diagnosis of PC, and 9% even in the process of monitoring the results of treatment. Due to the risks of ionizing radiation mentioned above, MRI should be used instead of CT scan to avoid radiation to assess PC [37].

The classic (intensive) concept of daily COB application recommends 23 h of daily wearing (except during sports, bathing, or showering) [3,21,24,26,29]. On the other side, a more practicable protocol recommends 8–12 h per day, with similarly favorable treatment results [10,14,20,25]. Other schemes are used as well, i.e., with daily time intervals between the former two [23]. No consensus and no uniform bracing protocol exist [20]. In addition, it has been reported that many patients do not follow the clinically prescribed treatment (e.g., applied pressure, usage time) [15]. The survey reveals comparative findings between CC and FMF in the sense of significantly shorter physician-recommended times of daily COB application in CC group. The patient-confirmed times of daily COB wearing were significantly shorter in CC versus FMF group as well. The most important finding is the confirmation of the assumption that the actual wearing time of COB is significantly lower than the physician recommended in both groups (CC and FMF). The analysis of Wahba et al. gives a key factor in the improvement of patient adherence to COB. His study showed that brace usage < 12 h/day is associated with higher patient compliance with a similar time to correction and success rate in comparison to the more rigorous protocol [20]. In this context, intensive therapeutic regimens > 12 h are to be questioned.

It remains difficult to define the success of COB because aesthetic self-assessment is very subjective. A well-described definition of a successfully corrected chest wall is lacking in all available studies. De Beer et al. in a multicenter review study postulate a concept of “correction” and a “retainer” mode, used by many centers [26]. Usually, the endpoint of the retainer phase is based on the subjective assessment by the surgeon [20] or depends on the judgment of the treating physician and the patient and his/her family [27]. Conversely, none of the available studies analyzed the decision when to switch from the correction phase to a maintenance phase. Our study confirmed that the success of COB is based mainly on patients’ subjective satisfaction with the result, and less frequent on objective clinical anthropometry or 3D surface scanning. The same indicators were used for decisions concerning the end of the treatment. In the future, it is assumed that greater emphasis will be placed on shared decision making about changing the treatment mode and ending treatment using the 3D camera technique and derivative graphics for objective measurements [27].

The overall treatment duration is another not-so-precisely predefined parameter, with highly variable values in the literature. Most studies indicate the total time of COB is in the interval of 6–12 months [3,21,23,38]. This contrasts with analyses indicating a significantly longer period (14–24 months) [25,27,39]. The study of Martinez-Ferro et al. offered an interesting finding, that even with a less intensive treatment protocol, the average duration of treatment was 7 months [10]. Our results show that the average duration of treatment was significantly longer with CC than with FMF (14.0 ± 4.3 vs. 11.0 ± 4.0 months). At the same time, however, the start of CC treatment was in significantly older children than in the case of FMF (9.5 ± 1.6 years vs. 7.5 ± 1.1 years) and the duration of daily application (physician-recommended and also patient-confirmed) was shorter in the CC group compared to the FMF group (14.3 ± 3.7 vs. 16.4 ± 4.0 h and 10.4 ± 3.8 vs. 14.8 ± 5.5 h, respectively).

Our study revealed that the patients were followed-up until the end of growth in 82%. This is a pleasingly high number, in the context of rather sporadic literary data. Shang et al. conducted follow-ups for at least 3 months and up to 3 years after finishing COB [23], de Beer et al. mentioned post-treatment follow-ups every 6 months until patients reached the age of 18 or stopped growing [26]. The low percentage of relapses described in the literature is noteworthy: 1.5–2.6%, according to another study conducted by de Beer et al. enrolling 740 patients [27]. Typically, the recurrence of sternal protrusion is associated with ongoing pubertal growth and responds to re-initiation of active bracing [25]. In contrast, in our study, where almost all departments report follow-up until the end of growth, 1/3 of the workplaces did not report recurrences, whereas 61% reported recurrence rates varying from 10 to 30% of patients, and even a 5% recurrence rate >50%.

Patient compliance has been reported as a key factor of successful treatment [15]. The literature highlights high levels of treatment abandonment, from 30 to 43% [16,22,25]. On one hand, a combination of negative factors such as the presence of pain, skin problems, shame, and discomfort were identified as significant predictors of non-compliance [15,19,21,26]; on the other hand, initial success of the compression period was a strong predictor of compliance [19,21]. Larger studies indicate better compliance: Shang et al. analyzed 664 patients who obtained satisfactory chest appearance through COB, with a success rate of up to 84% [23], Martinez-Ferro et al. with a group of 208 patients accomplished a total success rate of 86.6% [10]. Our study indicates a high (>80%) compliance rate in 74% of responses, despite a relatively high incidence of the complications associated with COB wearing (pain in 35%, local skin rashes in 26%, and itching in 19%).

## 5. Limitations

When interpreting the results of this survey it should be kept in mind that the response rate and related sample size of this study was rather small. We assume that this could be due to several facts. Some colleagues may not have answered because of: (i) little or no experience with COB, (ii) surgical approach preference (Ravitch or Abramson procedures and their modification), (iii) several email addresses belonging to the same institution, while only the head of department or consultant replied, (iv) heterogeneity of CWIG members (pediatric surgeons, thoracic surgeons, orthopedists and plastic surgeons, physiotherapists, physiologists, nurses), (v) some email addresses not being current, and (vi) other reasons unknown to us. In addition, web-based questionnaire surveys have several known limitations such as self-reporting bias, but are nevertheless an accepted way of investigating tendencies [40].

## 6. Conclusions

Although CT remains a very common imaging modality for the assessment of PC, clinicians need to be aware of the risk of ionizing radiation, especially in young patients. COB appears to be the first-line treatment for the majority of PC patients. Compliance with COB is high. As the main problems are a significant difference between the recommended and actual daily application time of COB and a high recurrence rate of PC during puberty, monitoring devices and standardization of COB practice are essential to improve the effectiveness of PC treatment. Follow-up until the end of puberty is strongly recommended.

## Figures and Tables

**Figure 1 children-11-00470-f001:**
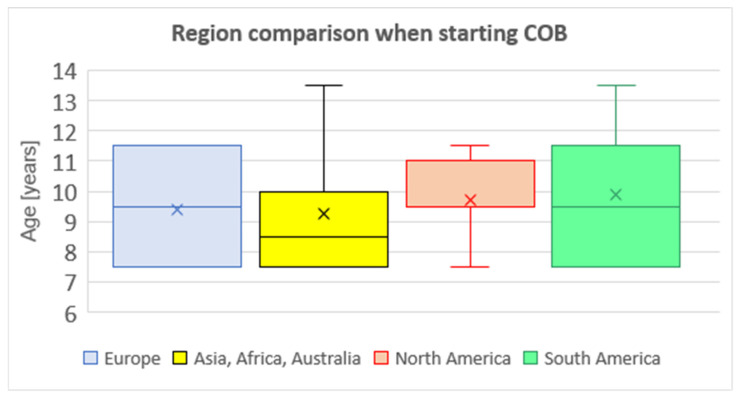
Region comparison when starting COB (compressive orthotic bracing).

**Figure 2 children-11-00470-f002:**
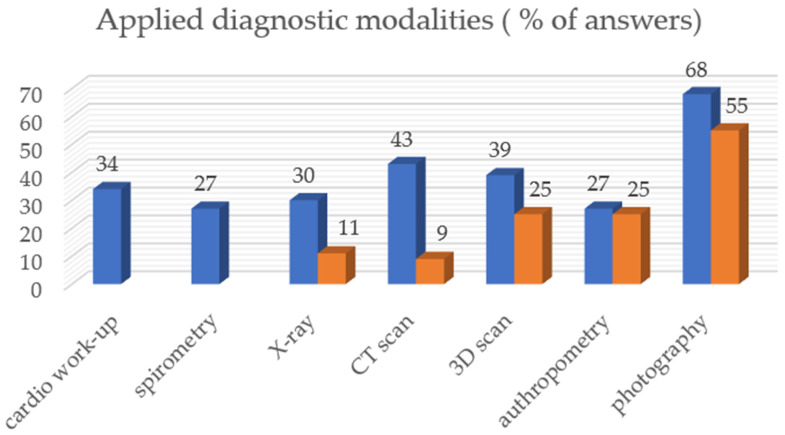
Diagnostic modalities employment during diagnostics and treatment.

**Figure 3 children-11-00470-f003:**
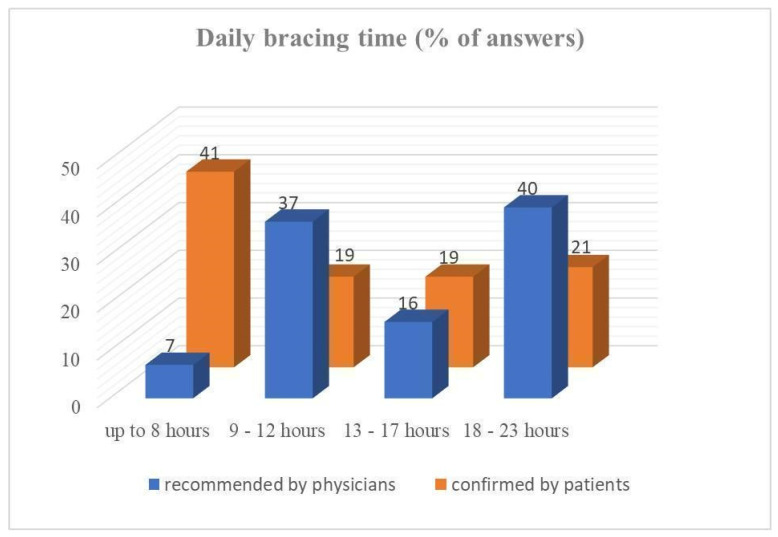
Daily bracing time of compressive orthotic bracing–recommended by physicians vs. confirmed by patients.

**Figure 4 children-11-00470-f004:**
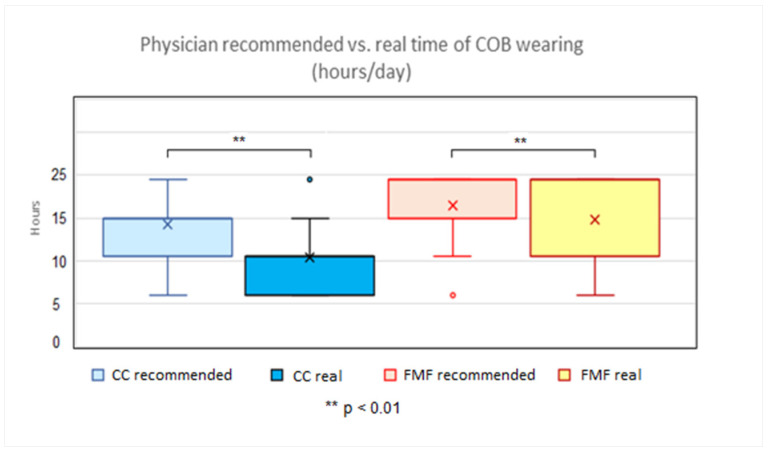
Physician-recommended vs. patient-confirmed time of compressive orthotic brace daily wearing, comparison of CC (compressive orthotic brace) and FMF (Fraire Martinez-Ferro dynamic compression system) group.

**Figure 5 children-11-00470-f005:**
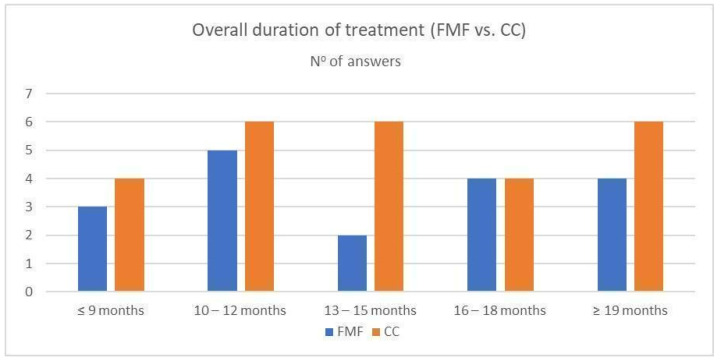
Overall duration of compressive orthotic bracing in CC (conventional compressive brace) and FMF (Fraire Martinez-Ferro dynamic compression system) group.

**Figure 6 children-11-00470-f006:**
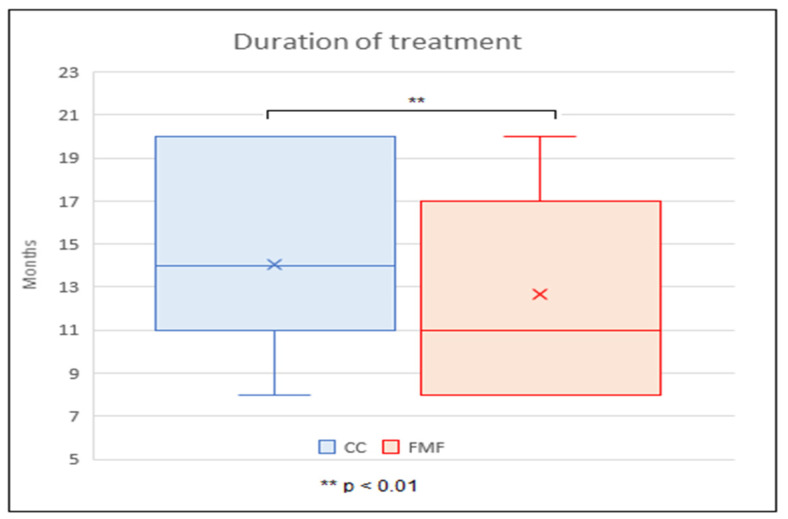
Comparison of compressive orthotic bracing duration between CC (conventional compressive brace) and FMF (Fraire Martinez-Ferro dynamic compression system) group.

**Figure 7 children-11-00470-f007:**
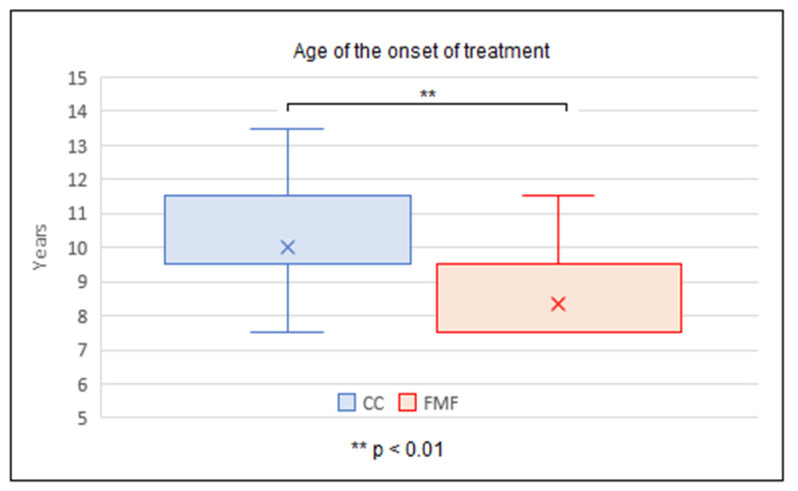
Age comparison at the onset of compressive orthotic bracing between CC (conventional compressive brace) and FMF (Fraire Martinez-Ferro dynamic compression system) group.

**Figure 8 children-11-00470-f008:**
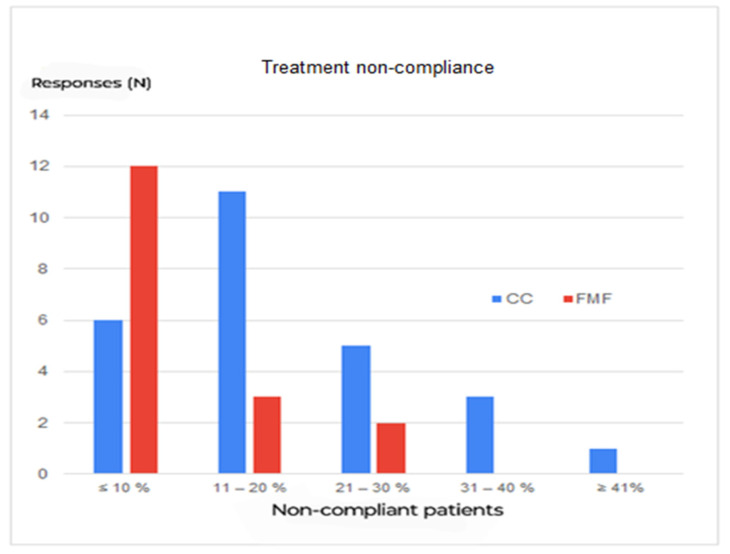
Treatment non-compliance in CC (conventional compressive brace) vs. FMF (Fraire Martinez-Ferro dynamic compression system) group.

**Table 1 children-11-00470-t001:** The experience of the participating departments.

Departments Experience	
Question	Answers N (%)
**Number of new patients with pectus carinatum per year**	
≤25	15 (34)
26–50	16 (37)
51–75	5 (11)
76–100	4 (9)
≥101	4 (9)
**What is your overall experience with PC external bracing?**	
≤3 years	5 (11)
4–6 years	11 (25)
7–9 years	8 (18)
≥10 years	20 (46)
**How many patients have you treated with COB successfully so far?**	
≤50	15 (34)
51–100	9 (20)
101–150	7 (16)
151–200	2 (5)
≥201	11 (25)

## Data Availability

Data are contained within the article and Appendix A.

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
