# Peer review of "Bracing of Pectus Carinatum in Children: Current Practices"

_children, 2024, doi:10.3390/children11040470_

Round 1

Reviewer 1 Report

Comments and Suggestions for Authors

The authors present the results of a survey sent to members of the Chest Wall International Group, where only 14% responded, so the results of the study have to be analysed with great caution, as it is a small percentage of the total number of professionals who treat this pathology.

The manuscript is adequately well structured, with good images and graphics. The references are up to date. I would change the presentation of the results, stressing the limited capacity to extrapolate them, due to the very low response rate obtained in the survey.

detailed comments:

  1. What is the main question addressed by the research? The authors present the results of a survey sent to members of the Chest Wall International Group. 2. What parts do you consider original or relevant for the field? What specific gap in the field does the paper address? The main advantage of this manuscript is that it is the first survey of members of this chest wall society on the management of pectus carinatum, exploring the different orthopaedic prostheses available, as well as patients' adherence to treatment. What does it add to the subject area compared with other published material? The article has several strengths such as the exploration of the different types of pectus carinatum management in different countries around the world, which are grouped into continents. It allows comparison of the management of patients with pectus carinatum in different parts of the world. 4. What specific improvements should the authors consider regarding the methodology? What further controls should be considered? The methodology is adequate, but the main problem is the low adherence to the survey response. 5. Please describe how the conclusions are or are not consistent with the evidence and arguments presented. Please also indicate if all main questions posed were addressed and by which specific experiments. The main drawback or limitation of this article is that only 14% responded, so the results of the study have to be analysed with great caution, as it is a small percentage of the total number of professionals who treat this pathology. 6. Are the references appropriate? References 13 and 15 are book chapters written by the authors of the article, they are articles not written in English, and therefore, difficult to read for the rest of the readers. The references should be changed to avoid self-citation.

7. Please include any additional comments on the tables and figures and quality of the data. The quality of the figures is adequate. However, figures 2, 3 and 5 do not appear in the manuscript, although the figure legends are in the manuscript.

Comments on the Quality of English Language

Minor editing of English language required

Author Response

The authors present the results of a survey sent to members of the Chest Wall International Group, where only 14% responded, so the results of the study have to be analysed with great caution, as it is a small percentage of the total number of professionals who treat this pathology.

ANSWER:

                We agree with the comment and we realize, that the response rate of 14% is low, however, we find more valuable the institutional response rate, which is rather higher (22%). CWIG is a heterogenous society of different specialists:  the ones who contribute to diagnostics, an indication of COB treatment and follow-up (paediatric surgeons, thoracic surgeons and plastic surgeons); the others are supportive specialists (physiotherapists, physiologists, nurses etc.), who probably did not answer the questionnaire. We could expect the answers from the first mentioned group, but unfortunately from the email addresses of CWIG members (total number of 313 members), we are not capable of selecting surgical specialists who participate in the follow-up of patients.  Analysis of email addresses showed the fact, that individual departments have various numbers of CWIG members (at least one member, maximum of seven members). We have to underline the fact, that the majority of answers from individual institutions come from either the head of department or consultants which determines institutional response rate as more relevant.

               The most important reason for the low response rate, in our opinion, is the fact that there are institutions which prefer surgical treatment of PC as a method of choice (Ravitch or Abramson procedures and their modification). Based on this and on the fact that we cannot recognise the part of CWIG members, who prefer the surgical method, we can suppose that a substantial part of CWIG members did not respond to the questionnaire due to the reason of their surgical preference.

Thank you for this important comment. However, the relatively low response rate of 14.1% must be interpreted with caution. Of the 313 colleagues contacted, the most important centres responded, including high-volume centres in Argentina, Belgium, Brazil, England, France, Italy, the Netherlands, South Africa, Spain, Switzerland and the USA. The centres in these countries are also those with the most experience in using the FMF.  The total number of included patients extrapolated from these centres was almost 4750.

This small number of responses can therefore be interpreted as a powerful statement

The manuscript is adequately well structured, with good images and graphics. The references are up to date. I would change the presentation of the results, stressing the limited capacity to extrapolate them, due to the very low response rate obtained in the survey.

ANSWER:

We have changed the presentation of the results considering the low response rate in the corrected version of the manuscript.

detailed comments:

  1. What is the main question addressed by the research?

The authors present the results of a survey sent to members of the Chest Wall International Group.

  1. What parts do you consider original or relevant for the field? What specific gap in the field does the paper address?

The main advantage of this manuscript is that it is the first survey of members of this chest wall society on the management of pectus carinatum, exploring the different orthopaedic prostheses available, as well as patients' adherence to treatment.

What does it add to the subject area compared with other published material?

The article has several strengths such as the exploration of the different types of pectus carinatum management in different countries around the world, which are grouped into continents. It allows comparison of the management of patients with pectus carinatum in different parts of the world.

  1. What specific improvements should the authors consider regarding the methodology? What further controls should be considered?

The methodology is adequate, but the main problem is the low adherence to the survey response.

ANSWER:

We are aware of the limitations of the study due to the low response rate, we have explained them sooner.

  1. Please describe how the conclusions are or are not consistent with the evidence and arguments presented. Please also indicate if all main questions posed were addressed and by which specific experiments.

The main drawback or limitation of this article is that only 14% responded, so the results of the study have to be analysed with great caution, as it is a small percentage of the total number of professionals who treat this pathology.

  1. Are the references appropriate?

References 13 and 15 are book chapters written by the authors of the article, they are articles not written in English, and therefore, difficult to read for the rest of the readers. The references should be changed to avoid self-citation.

ANSWER: 

Thank you for that comment, we have already changed references to international articles to avoid self-citation.

  1. Please include any additional comments on the tables and figures and quality of the data.

The quality of the figures is adequate. However, figures 2, 3 and 5 do not appear in the manuscript, although the figure legends are in the manuscript.

ANSWER:

We apologize for the inconvenience. We have completed the missing figures in the definitive version of the manuscript.

Reviewer 2 Report

Comments and Suggestions for Authors

This is an important topic to study and the authors should be commenced for their efforts to illuminate this area. 

Was this questionnaire validated?

If yes, which institution validated this questionnaire?

The manuscript is written in a way that when a reader reads the results and the discussion sections, the reader gets the impression that this study is based on analysis of 4750 PC patients! In fact, it is not possible to know how many patients are really treated by the people answering the 30 questions in the time period of 10 minutes. The answered questions only give the subjective impression of the people who answered the questionnaire. Therefore, the authors must write this manuscript with great caution, and always refer to the results as answers from the questionnaire.

Also, the questions 15 and 16 refer to the method of compressive orthotic bracing (COB), which is furthermore quantified with patient numbers and statistical analysis comparing the different answers from colleagues filling the questionnaire about their impressions, like for example, did their patients wear the brace as recommended or for a shorter period of time?! How can this question be answered "correctly" without careful analysis of own patients? This kind of analysis is to my opinion not appropriate and I would not consider seriously these results. It would be different if these results would be based on published data, as in the case of meta-analysis, but not in this way. 

Findings like, usage of CT for diagnostic purposes, and various differences in the approach to PC treatment are helpful and informative. But statistical analysis showing that treatment with FMF shows a significantly faster response than treatment with CC, based on this questionnaire is completely inacceptable. 

Remark of the authors about the limitations of the study are fine, but the manuscript should be written in the way that these limitations are seen throughout the manuscript, as self-reporting bias may be huge in this area.   

Also, every table or figure should be able to stand alone, therefore all abbreviations should be indicated in the Legends or in a box. 

Finally, COB is strongly dependent on the rigidity of the thoracic wall. In the past, a threshhold of 10 Kg (100 N) was used to assess the indication for surgical or conservative approach to PC. The examination was performed lying down in a supine position. Martinez-Ferro and colleagues introduced the measurement based on pounds per square inch (PSI) in standing position. It is somewhat strange that the authors where not interested to assess the opinion of the colleagues on the influence of the rigidity of the chest wall on the success of COB treatment. 

Author Response

Was this questionnaire validated? If yes, which institution validated this questionnaire?

ANSWER:

The questionnaire was prepared by the cooperation of two European institutions and approved by the President of CWIG.

The manuscript is written in a way that when a reader reads the results and the discussion sections, the reader gets the impression that this study is based on analysis of 4750 PC patients! In fact, it is not possible to know how many patients are really treated by the people answering the 30 questions in the time period of 10 minutes. The answered questions only give the subjective impression of the people who answered the questionnaire. Therefore, the authors must write this manuscript with great caution, and always refer to the results as answers from the questionnaire.

ANSWER:

The manuscript presents the evaluation of the questionnaire, sent to the members of CWIG and thus can't provide the patients' analysis, nevertheless, it offers the experience of different institutions with the treatment of PC - the total number of 4750 patients stands for the long-term experience of PC treatment in 44 institutions (high number of treated patients at institutions which are mainly pectus centres). The number of patients was calculated as the weighted sum of sorted data using the middle value of each class, therefore, the resulting number can be taken as a reliable one (although it seems to be high). The questions in the questionnaire were prepared consistently to obtain relevant data from the experience of individual institutions based on their standardized protocols.

Also, the questions 15 and 16 refer to the method of compressive orthotic bracing (COB), which is furthermore quantified with patient numbers and statistical analysis comparing the different answers from colleagues filling the questionnaire about their impressions, like for example, did their patients wear the brace as recommended or for a shorter period of time?! How can this question be answered "correctly" without careful analysis of own patients? This kind of analysis is to my opinion not appropriate and I would not consider seriously these results. It would be different if these results would be based on published data, as in the case of meta-analysis, but not in this way.

ANSWER:

Questions 15 and 16 were created to divide respondents into 2 basic groups: the first one that uses FMF and the second one with the use of CC system. More specified questions were not incorporated into the questionnaire to avoid an extensive pool of information. 

Questions 17 and 18 regarding recommended and real-time orthosis wearing were specified with definite time intervals and we suppose that the answers were based on institutional algorithms of COB treatment and that all respondents have their own analysis of treated patients.

According to our knowledge, the literature does not comprise meta-analysis regarding the comparison of FMF and CC orthotic systems to date. We do not insist on the creation of definite guidelines for the treatment of PC, questionnaire was formed to put insight into the therapeutic strategies at individual institutions. However, the authors hope to continue with the multicentric study based on the presented data.

Findings like, usage of CT for diagnostic purposes, and various differences in the approach to PC treatment are helpful and informative. But statistical analysis showing that treatment with FMF shows a significantly faster response than treatment with CC, based on this questionnaire is completely inacceptable.

ANSWER:

The authors compared two modes of COB treatment (FMF and CC)  based on answers from questions 15,16. The average duration of treatment in both groups was statistically analysed according to answers to question 23. The average duration of treatment  (FMF vs CC group) was calculated as the weighted sum of sorted data using the middle value of each class, therefore, the resulting number can be taken as a reliable one.

Remark of the authors about the limitations of the study are fine, but the manuscript should be written in the way that these limitations are seen throughout the manuscript, as self-reporting bias may be huge in this area.  

ANSWER:

We are aware of limitations and we could underline them in the text.

Also, every table or figure should be able to stand alone, therefore all abbreviations should be indicated in the Legends or in a box.

ANSWER:

We agree, we will add the legends with abbreviations

Finally, COB is strongly dependent on the rigidity of the thoracic wall. In the past, a threshhold of 10 Kg (100 N) was used to assess the indication for surgical or conservative approach to PC. The examination was performed lying down in a supine position. Martinez-Ferro and colleagues introduced the measurement based on pounds per square inch (PSI) in standing position. It is somewhat strange that the authors where not interested to assess the opinion of the colleagues on the influence of the rigidity of the chest wall on the success of COB treatment.

 ANSWER:

The authors decided to put 30 questions to avoid the extensive amount of time needed to fulfil the answers. We realize the importance of the reviewer's comment regarding the rigidity of the chest wall, however, for the complex analysis of the subject we would need to add at least three questions more. The questionnaire was prepared with the hope to initiate a multicentric prospective/retrospective study regarding conservative treatment of PC and this would be open for more detailed analysis.

Reviewer 3 Report

Comments and Suggestions for Authors

The study provides valuable insights into the practices of Chest Wall International Group (CWIG) members regarding compressive orthotic bracing (COB) in children with pectus carinatum (PC). Further elaboration on the challenges faced in standardizing COB practices and the proposed recommendations for enhancing its effectiveness during puberty would enhance the research's completeness. Overall, the study addresses an important topic within the field.

Author Response

The study provides valuable insights into the practices of Chest Wall International Group (CWIG) members regarding compressive orthotic bracing (COB) in children with pectus carinatum (PC). Further elaboration on the challenges faced in standardizing COB practices and the proposed recommendations for enhancing its effectiveness during puberty would enhance the research's completeness. Overall, the study addresses an important topic within the field.

Thank you for your valuable review and comment, really appreciated!

Round 2

Reviewer 2 Report

Comments and Suggestions for Authors

I acknowledge the efforts made by the authors to improve the manuscript. 

I also think that content of this manuscript should serve to stimulate further research on this topic as the authors mentioned it in their comments. I think this needs to be underlined even more strongly in the manuscript. 

Finally, I dont think that on the basis of this questionairre statistical analysis should be made and conclusions should be drown concerning which  COB gives better results, FMF or CC. I find this inappropriate and potentially misleading. 

Author Response

Reply to reviewer comments:

  • I acknowledge the efforts made by the authors to improve the manuscript. 

    Thank you for your kind reply, really appreciated

  • I also think that content of this manuscript should serve to stimulate further research on this topic as the authors mentioned it in their comments. I think this needs to be underlined even more strongly in the manuscript.

    Thank you for your comment. We tried to comment on that issue in the “discussion” paragraph (starting line 224 onwards)
  • Finally, I dont think that on the basis of this questionnaire statistical analysis should be made and conclusions should be drown concerning which COB gives better results, FMF or CC. I find this inappropriate and potentially misleading.

    The authors are aware that they are processing questions from an anonymous questionnaire. Therefore, the correctness of the answers is neither verifiable nor reproducible. However, it is a completely common procedure that questionnaire data, which are of a quantitative nature, are evaluated using statistical methods. The questionnaire used contained several questions of this nature. The same statistical procedures were used for the same type of questions.

    Nevertheless, we added a comment in the “discussion” line 290 onwards.